# Integrated Amino Acids and Transcriptome Analysis Reveals Arginine Transporter SLC7A2 Is a Novel Regulator of Myogenic Differentiation

**DOI:** 10.3390/ijms25010095

**Published:** 2023-12-20

**Authors:** Tiane Huang, Jing Zhou, Benhui Wang, Xiang Wang, Wanli Xiao, Mengqi Yang, Yan Liu, Qiquan Wang, Yang Xiang, Xinqiang Lan

**Affiliations:** Metabolic Control and Aging—Jiangxi Key Laboratory of Human Aging, Human Aging Research Institute (HARI), School of Life Science, Nanchang University, Nanchang 330031, China; huangtiane@ncdx.wecom.work (T.H.); jingzhou2303@163.com (J.Z.); wbhxm8@163.com (B.W.); wx18162110448@163.com (X.W.); 352428819003@email.ncu.edu.cn (W.X.); yang15774544569@163.com (M.Y.); liuyan09192022@163.com (Y.L.); wangqiquan@ncu.edu.cn (Q.W.)

**Keywords:** myogenic differentiation, amino acids, transporter, SLC7A2, sarcopenia

## Abstract

Skeletal muscle differentiation is a precisely coordinated process. While many of the molecular details of myogenesis have been investigated extensively, the dynamic changes and functions of amino acids and related transporters remain unknown. In this study, we conducted a comprehensive analysis of amino acid levels during different time points of C2C12 myoblast differentiation using high-performance liquid chromatography (HPLC). Our findings revealed that the levels of most amino acids exhibited an initial increase at the onset of differentiation, reaching their peak typically on the fourth or sixth day, followed by a decline on the eighth day. Particularly, arginine and branched-chain amino acids showed a prominent increase during this period. Furthermore, we used RNA-seq analysis to show that the gene encoding the arginine transporter, *Slc7a2*, is significantly upregulated during differentiation. Knockdown of *Slc7a2* gene expression resulted in a significant decrease in myoblast proliferation and led to a reduction in the expression levels of crucial myogenic regulatory factors, hindering the process of myoblast differentiation, fusion, and subsequent myotube formation. Lastly, we assessed the expression level of *Slc7a2* during aging in humans and mice and found an upregulation of *Slc7a2* expression during the aging process. These findings collectively suggest that the arginine transporter SLC7A2 plays a critical role in facilitating skeletal muscle differentiation and may hold potential as a therapeutic target for sarcopenia.

## 1. Introduction

Skeletal muscle, which accounts for approximately 40% of body weight, plays a vital role in human activities and movements [1]. The size of muscle fibers is influenced by various factors including exercise, nutrition, and aging [2,3,4]. Muscle fiber growth is dependent on a process known as myogenesis, which involves the proliferation of satellite cells. These satellite cells give rise to myoblasts that undergo multiplication, differentiation, and fusion to form muscle fibers [5]. This process occurs during both normal muscle development and muscle regeneration following injury. The progression of myogenesis is controlled by the spatiotemporal expression of various myogenic regulatory factors and transcription factors such as MyoD and Myogenin [6]. These factors regulate the withdrawal of proliferating myoblasts from the cell cycle, their elongation, adhesion, and fusion into multinucleated myotubes. Myotubes exhibit distinct morphology, structure, and function compared to myoblasts, with increased expression of proteins associated with muscle contraction. During myogenic differentiation, new proteins are synthesized while others are selectively degraded. Proper protein expression is crucial for the orderly progression of myogenesis, and controlled protein degradation occurs through various mechanisms including cathepsins, calpains, caspases, and the ubiquitin-proteasome system (UPS) [7].

Amino acids play a critical role in protein synthesis as they serve as the fundamental building blocks of proteins [8]. Recent research has revealed that amino acids also function as signaling molecules, regulating the intricate process of protein synthesis. One prominent example of this is the mammalian target of rapamycin (mTOR) pathway [9,10], which detects the presence of specific amino acids, such as leucine, and triggers protein synthesis accordingly [11]. In addition to their involvement in protein synthesis, amino acids also play a vital role in various metabolic pathways and energy production [12]. They can be converted into glucose through a process known as gluconeogenesis, providing a valuable source of energy when carbohydrate stores are depleted [13]. Moreover, certain amino acids can be oxidized to generate energy during times of increased energy demands [14]. These findings emphasize the importance of maintaining an appropriate concentration of amino acids for various biological processes and overall health [15].

Amino acid transporters are integral membrane proteins that play a crucial role in facilitating the transport of amino acids across cell membranes [16]. These transporters are essential for maintaining amino acid homeostasis and are involved in various physiological processes. They are classified into different families based on their structural and functional characteristics, such as the SLC7, SLC1, and SLC6 families [17,18]. The SLC7 family, also known as the cationic amino acid transporter family, is responsible for transporting cationic amino acids like arginine, lysine, and ornithine [19]. These transporters are essential for protein synthesis, cellular signaling, and nutrient uptake regulation [20,21]. The SLC1 family, also known as the excitatory amino acid transporter family, primarily transports glutamate and aspartate [22]. These amino acids serve as important neurotransmitters in the central nervous system and play a critical role in synaptic transmission and excitatory signaling [23]. The SLC6 family, also known as the neurotransmitter transporter family, transports amino acids such as glycine, proline, and alanine [24]. These transporters are involved in the reuptake of neurotransmitters from the synaptic cleft, regulating their concentration and terminating signaling [25]. The expression and activity of amino acid transporters are tightly regulated to ensure proper amino acid uptake and distribution. Various factors, including hormonal signals, nutrient availability, and cellular energy status, can modulate the expression and activity of these transporters [26].

Although the functions of amino acids and their transporters have been extensively studied in various physiological aspects such as immune regulation and neuronal signaling, there is limited research on the dynamic changes of amino acids and the expression of related transporters during muscle differentiation. Therefore, the objective of this study is to investigate the role of amino acids and amino acid transporters in skeletal muscle differentiation.

## 2. Results

### 2.1. Amino Acids Levels Exhibit Distinct Changes during C2C12 Myoblast Differentiation

To investigate the dynamic changes in amino acid levels during differentiation, we utilized the C2C12 myoblast line, a well-established in vitro model for studying skeletal muscle differentiation [27]. Amino acids were derivatized using the DNFB pre-column derivatization method and separated and identified by HPLC with UV absorption detection. The levels of amino acids were analyzed during the 8-day differentiation period of C2C12 cells (Figure 1A). The results revealed distinct changes in amino acid levels throughout the differentiation process (Figure 1B). Notably, arginine exhibited a significant increase, reaching its peak on the fourth day of differentiation, suggesting its potential role in muscle differentiation at this specific time point. Branched-chain amino acids (BCAAs), including leucine, isoleucine, and valine, gradually increased and peaked on the sixth day, indicating their significance in muscle differentiation. Glycine, alanine, proline, serine, and threonine initially increased and then decreased, suggesting their involvement in early stages of muscle differentiation. In contrast, methionine, glutamic acid, aspartic acid, phenylalanine, and tryptophan showed an increase after differentiation, reaching their peak on the fourth day, and subsequently decreasing. Tyrosine and lysine displayed their highest levels on the fourth and sixth day, respectively, suggesting their potential involvement in muscle differentiation at specific time points. Histidine exhibited peak content on both day 0 and day 6, indicating its importance in the early and later stages of differentiation. These findings highlight the distinct and dynamic changes in amino acid levels throughout the entire differentiation process, emphasizing the potential roles of specific amino acids at different differentiation stages.

### 2.2. Amino Acid Metabolism Related Genes Are Enriched during C2C12 Myoblast Differentiation

To determine the gene expression profile during C2C12 myoblast differentiation, RNA sequencing analysis was performed on myoblasts at different time points (0, 2, 4, 6, 8 days). Firstly, we conducted cluster analysis of genes and samples for RNA-seq data and found that the samples were correlated (Figure 2A). We performed a pattern analysis using STEM (Version 1.3.11) software, which has been widely used for in-depth analysis of time-course gene expression data. The results revealed 14 significant expression patterns during C2C12 differentiation (*p* < 0.001). They were generally classified into 4 categories: (1) increasing (module 49) or decreasing (module 0) during the entire time course of differentiation; (2) dramatically increasing (module 45) or decreasing (module 6) when differentiation started; (3) increasing (module 30) or decreasing (module 25) from 4 days post differentiation; and (4) having the highest or lowest expression level during the differentiation process (module 46, 34, 43, 29, 5, 19, 33, 48) (Figure 2B). To elucidate the functional consequences of the gene’s expression change in different modules, we carried out KEGG pathway enrichment in different modules, as described in the Methods. There are 574 genes in modules 0 and 49, they were enriched in “DNA replication”, “Cell cycle”, “Fanconi anemia pathway” and other pathways (Figure 2C). These pathways are generally related to cell proliferation and represent the cells that changed their state from proliferation to differentiation. We further combined module 6 and module 45 for KEGG enrichment analysis, and the results showed that 40 pathways were enriched (Figure 2D). When considering the metabolism category of KEGG pathways, 9 of them belong to “amino acid metabolism” and “Metabolism of other amino acids” (Figure 2E,F). The following “amino acid metabolism” pathways are included: “Cysteine and methionine metabolism”, “Valine, leucine, and isoleucine degradation”, “Arginine and proline metabolism”, “Lysine degradation”, “Glycine, serine and threonine metabolism”, “Valine, leucine and isoleucine biosynthesis”, “Histidine metabolism”. These results indicated the pivotal role of amino acid metabolism in myoblast differentiation.

### 2.3. Arginine Transporter SLC7A2 Is Upregulated during Myogenic Differentiation

The functionality of amino acids is primarily dependent on their transporters. To investigate changes in amino acid transporters during C2C12 differentiation, we analyzed the expression of 29 reported genes encoding amino acid transporters in C2C12 cells. By comparing the differential gene expression profiles on day 0 and day 2, we identified six significantly upregulated amino acid transporter-related genes: *Slc7a2*, *Slc1a3*, *Slc7a7*, *Slc25a13*, *Slc43a2*, and *Slc6a7*. Notably, *Slc7a2* and *Slc1a3* exhibited the highest fold change (Figure 3A). To gain further insights, we examined the expression patterns of *Slc1a3* and *Slc7a2* in various tissues using the GTEx dataset. Available online: https://www.proteinatlas.org/ENSG00000003989-SLC7A2/tissue (accessed on 23 September 2022). Slc1a3 was found to be highly expressed in human kidney, pancreas, and intestinal tissues, but had low expression in skeletal muscle tissues. Conversely, *Slc7a2* showed high expression in skeletal muscle and liver tissue. Additionally, given the function of *Slc1a3* involved in proliferation in cancer and stem cell have been reported [28], while *Slc7a2* was rarely mentioned. These findings suggest a potential role for *Slc7a2* in myoblast differentiation.

The above results also revealed the level of arginine exhibited a significant increase on the fourth day of differentiation (Figure 1B). Therefore, we focused our research on investigating the impact of *Slc7a2* on myoblast proliferation and differentiation. To explore the involvement of *Slc7a2* in C2C12 myoblast differentiation, we further analyzed its expression using qPCR and Western blotting. Our results revealed a gradual upregulation of *Slc7a2* at both the mRNA and protein levels during the differentiation period (Figure 3B,C). These findings provide evidence for the involvement of *Slc7a2* in the myogenic differentiation of C2C12 myoblasts.

### 2.4. Knockdown of SLC7A2 Inhibits C2C12 Myoblast Proliferation

To further investigate the potential role of SLC7A2 in cell proliferation of myoblasts, we transfected C2C12 cells with *Slc7a2* siRNA. After allowing the cells to grow for 48 h, we observed the successful suppression of SLC7A2 mRNA and protein expression following si-SLC7A2 transfection (Figure 4A,B). Knockdown of SLC7A2 significantly reduced the concentration of arginine in C2C12 cells (Figure 4C) and the concentration changes of other amino acids are shown in Appendix A. To assess the proliferative capacity, we performed immunocytochemistry analysis using Ki67, a commonly used marker for proliferation experiments. The results demonstrated a significant reduction in the number of Ki67-positive cells upon SLC7A2 knockdown (Figure 4D). These findings indicate that SLC7A2 plays a crucial role in promoting cell proliferation in C2C12 myoblasts.

### 2.5. Knockdown of SLC7A2 Inhibits Myoblast Differentiation

To explore the role of SLC7A2 in myoblast differentiation, C2C12 myoblasts were transfected with si-SLC7A2 and then differentiated for 2 days. The differentiation status of C2C12 cells was assessed using MyHC immunofluorescence assay. We observed a decrease in the number of myotubes upon interference with SLC7A2 expression, indicating impaired differentiation (Figure 5A). Furthermore, interfering with SLC7A2 expression resulted in reduced mRNA expression levels of *Myogenin* and *MyoD*, key markers of myoblast differentiation (Figure 5B). qPCR analysis confirmed that the mRNA levels of type II (*Myh1* and *Myh4*) and type I (*Myh7*) muscle fiber markers were significantly decreased in the knockdown group compared to the control group, indicating a disruption in muscle fiber formation (Figure 5C). These findings suggest that SLC7A2 plays a crucial role in promoting myoblast differentiation and the formation of functional muscle fibers.

### 2.6. The Expression of SLC7A2 Is Upregulated during Aging

Sarcopenia, the decline in skeletal muscle mass and function associated with aging, is often attributed to a reduction in the number and activity of satellite cells [29,30]. The above results indicate that SLC7A2 plays a crucial role in the proliferation and differentiation of myoblasts. To further investigate the potential role of SLC7A2 in age-related sarcopenia, we analyzed the expression of SLC7A2 during the aging process in public databases for humans skeletal muscle, available online: http://gb.whu.edu.cn/ADEIP or http://geneyun.net/ADEIP (accessed on 30 July 2022), and mouse skeletal muscle, available online: https://sarcoatlas.scicore.unibas.ch/ (accessed on 22 September 2022), skeletal muscle. Our analysis revealed a gradual increase in the expression level of SLC7A2 in both human and mouse skeletal muscle with advancing age (Figure 6A,B). These findings suggest that SLC7A2 may relate to the age-related changes observed in skeletal muscle. To provide additional support, we performed mRNA analysis of the gastrocnemius muscles from 22-month-old mice using qPCR, which confirmed higher level of *Slc7a2* compared to 3-month-old mice (Figure 6C).

In summary, all of these findings indicated that the upregulation of SLC7A2 during the aging process may have a significant role in influencing the proliferation and differentiation of myoblasts, thereby playing an important role in combating the development of sarcopenia.

## 3. Discussion

Skeletal muscle differentiation is a complex and tightly regulated process, yet there is limited understanding of the dynamic changes and functional roles of amino acids and their associated transporters in this process. In this study, we conducted a thorough analysis of the involvement of amino acids and amino acid transporters in the process of skeletal muscle differentiation. To accomplish this, we employed HPLC to comprehensively examine the levels of amino acids at various stages of C2C12 myoblast differentiation. Our findings demonstrated notable fluctuations in amino acid levels throughout the process of differentiation, with an initial increase followed by a subsequent decline in most amino acids. To gain further insights into this phenomenon, we utilized RNA-seq analysis to examine the expression patterns of amino acid transporters. Notably, we observed a significant upregulation of *Slc7a2*, the gene responsible for encoding the arginine transporter, during differentiation. To assess the functional importance of *Slc7a2*, we conducted knockdown experiments and observed a considerable decrease in myoblast proliferation. Furthermore, the downregulation of *Slc7a2* resulted in reduced expression levels of crucial myogenic regulatory factors, thereby impairing myoblast differentiation, fusion, and myotube formation. To the best of our knowledge, this study represents the first comprehensive investigation into the dynamic changes in amino acids and their transporters during muscle differentiation, as well as their potential functional roles.

Our study provides insights into the alterations in amino acid levels during muscle differentiation. We observed distinct changes in the levels of various amino acids, highlighting their potential roles in regulating this process. Specifically, we found that arginine exhibited a significant increase on the fourth day of differentiation, suggesting its involvement in muscle differentiation at this specific time point. Additionally, the gradual increase and peak of branched-chain amino acids (BCAAs) such as leucine, isoleucine, and valine on the sixth day indicate their significance in muscle differentiation. Furthermore, other amino acids including glycine, alanine, proline, serine, and threonine demonstrated an initial increase followed by a subsequent decrease, suggesting their involvement in the early stages of muscle differentiation. The distinct changes in amino acid levels during myoblast differentiation suggest a complex metabolic reprogramming that occurs during this process [14,31]. In addition to act as the building blocks of proteins, amino acids also play diverse and essential roles in various biological processes, including protein synthesis, metabolic regulation, energy supply, neurotransmission, immune modulation, and cell signaling [32,33]. The specific roles of different amino acids at different time points indicate their importance in regulating various aspects of muscle differentiation, such as cell proliferation, protein synthesis, and energy metabolism [34,35]. Our RNA-seq analysis revealed significant expression patterns of metabolic-related genes during myoblast differentiation. KEGG pathway enrichment analysis indicated that these genes were enriched in pathways related to cell proliferation, lipid metabolism, and amino acid metabolism [14,36]. These findings collectively suggest a pivotal role of metabolic reprogramming, specifically amino acid metabolism, in myoblast differentiation. Further investigation into the specific mechanisms by which these amino acids and related metabolic pathways regulate muscle differentiation will provide deeper insights into the molecular processes underlying muscle development and regeneration.

RNA-seq analysis showed that there are six amino acid transporter-related genes that are significantly upregulated during myoblast differentiation. Among the upregulated amino acid transporter-related genes, *Slc7a2* showed the highest fold change. Knockdown of SLC7A2 inhibited myoblast proliferation and differentiation, indicating its crucial role in promoting cell proliferation and muscle fiber formation. The upregulation of SLC7A2 during myoblast differentiation and its crucial role in promoting cell proliferation and muscle fiber formation provide further evidence for the importance of amino acid transporters in muscle development [37]. Sarcopenia is characterized by a progressive loss of muscle mass and function, and the decline in the number and activity of satellite cells is thought to contribute to this process [38,39]. We also observed the increased expression of SLC7A2 in aged human and mice skeletal muscle, this suggests a potential link between altered amino acid metabolism and age-related muscle dysfunction [40]. Investigating the functional consequences of SLC7A2 dysregulation in aged muscle and its potential as a therapeutic target for preventing or reversing sarcopenia will be a warranted subject.

In conclusion, our study provides novel insights into the dynamic changes in amino acid levels and the involvement of amino acid transporters, particularly SLC7A2, in myoblast differentiation. These findings highlight the intricate interplay between amino acids and muscle development and suggest potential therapeutic strategies for promoting muscle regeneration and preventing age-related muscle decline.

## 4. Materials and Methods

### 4.1. Cell Culture

Mouse C2C12 myoblasts were cultured under the standard conditions of 37 °C and a 5% CO_2_ humidified atmosphere in high-glucose Dulbecco’s Modified Eagle Medium (DMEM; Biological Industries, Cromwell, CT, USA) that was supplemented with 10% fetal bovine serum (Biological Industries) without adding any antibiotics. Subsequently, the C2C12 cells were switched to a differentiation medium (DM; Gibco, Billings, MT, USA) containing 2% horse serum (Bmassay, Beijing, China). Most myoblasts formed mature myotubes in 6 days.

### 4.2. RNA Sequencing

For C2C12 myoblast RNA-seq, C2C12 cultured in DM (90–100% confluence, cultured in differentiation medium for 2, 4, 6 days) were harvested and total RNA was extracted using RNAiso reagent (Takara, Kusatsu, Japan). Then, the RNA samples were sent to Majorbio Bio-pharm Technology Co., Ltd. (Shanghai, China) for RNA-seq.

### 4.3. Time-Course Differential Expression Gene Analysis

Time-course DEG analysis was performed according to the R package microarray Significant Profiles (maSigPro) (version 1.56.0). Briefly, to obtain the candidate genes, the least squares method was used to determine the coefficients of each independent variable, and the significance of the equation was evaluated according to the F-test. Then, stepwise regression was used to determine the best combination of independent variables and screen the significant genes, which led to the comprehensive time-course DEGs.

### 4.4. RNA Interference

For the siRNA knock-down experiments, C2C12 myoblasts were cultured in 12-well plates (5 × 105 cells). The cells in each well were transfected with 100 nM siRNAs using the Lipofectamine RNAiMAX Transfection Reagent (Invitrogen, Carlsbad, CA, USA) for 2 days, as per the manufacturer’s instruction. The oligonucleotide sequences for SLC7A2 siRNA were as follows: si-SLC7A1-1 (5′-AGACACAGCUUGGAGGGUATT-3′ for the sense strand and 5′-UACCCUCCAAGCUGUGUCUTT-3′ for the antisense strand); si-SLC7A2-2 (5′-CAAUGCAGCAACAGAAGAATT-3′ for the sense strand and 5′-UUCUUCUGUUGCUGCAUUGTT-3′ for the antisense strand).

### 4.5. Quantitative RT-PCR

Total RNA was extracted with RNAiso PLUS (TaKaRa, Kusatsu, Japan) and reverse transcribed by utilizing PrimeScript TM RT reagent Kit With gDNA Eraser (TaKaRa, Japan) according to the manufacturer’s instruction. Real-time qPCR was performed using the Realtime PCR Super mix with anti-Taq (Mei5 Biotechnology Co., Ltd., Beijing, China). The primer sequences were as follows Table 1.

### 4.6. Western Blot

The cells in the six-well plate were washed three times with pre-cooled phosphate-buffered saline (PBS; pH, 7.4), and the cells were lysed with 200 μL of protein RIPA Lysis Buffer per well. The total protein content was determined by Pierce™ BCA Protein Assay Kits (Thermo Fisher, Waltham, MA, USA, 23227) and NANODROP ONE OD260 nm. The collected protein samples were mixed with 2× SDS–PAGE Sample Loading Buffer, boiled, and centrifuged at 1000 rpm for 5 min. The processed protein sample (10 μg per well) was separated by 10% SDS–PAGE at 80 V in the stacking gel and 120V in the separating gel [41]. Proteins in the gel were transferred to a polyvinylidene fluoride (PVDF) membrane at 200 mA (Millipore, Billerica, MA, USA). The membrane was blocked with 3% bovine serum albumins (Bmassay, Beijing, China)/PBST for 2 h at room temperature, and then the membranes were incubated with primary antibodies: anti-SLC7A2 (1:1000, Abcam, Cambridge, UK, ab140831), MyH2 (1:1000, Santa Cruz Biotechnology, Dallas, TX, USA, sc-53095), β-actin (1:20,000, Proteintech, Rosemont, IL, USA, 81115-1-RR),, GAPDH (1:20,000, Proteintech, Rosemont, IL, USA, 10494-1-AP).

### 4.7. Immunofluorescence Assay

Cells on coverslips were fixed with methanol at −4 °C for 10 min and then washed with 1× PBS. cells were incubated for 1 h with PBST containing 1% BSA and incubated with diluted primary antibodies against MYH2 (1:200, Santa Cruz Biotechnology, sc-53095) and Ki67 (1:200; Abcam) for overnight at 4 °C. The cells were then rinsed three times with PBST and incubated with the corresponding FITC-conjugated secondary antibody (1:500; Biolegend, San Diego, CA, USA, 405305) for 2 h at room temperature. Next, the cells were rinsed three times with PBST and incubated in DAPI for 30 min to visualize the nuclei. The cells were again rinsed three times with PBS, and observation.

### 4.8. HPLC Method for Quantification of Intracellular Amino Acid Concentrations in C2C12 Cell

Sample extraction: C2C12 cells were trypsinized for 1 min, 3 mL DMEM (10% FBS) was added to stop the digestion, counted, centrifuged to remove the supernatant and weighed, washed twice with 1 × PBS solution, centrifuged at 600× *g* for 5 min each time, and the supernatant was discarded to retain the pellet. After the cell suspension was transferred to a 2 mL centrifuge tube, the total cell weight was weighed and recorded. The bottom was flicked to loosen the cells, 100 μL of ultrapure water and liquid nitrogen for 1 min were added, and sonicated in an ice bath for 2 min, repeated six times. 500 μL of acetonitrile pre-cooled at −20 °C was added, mixed well at −20 °C for 2 h, and centrifuged at 15,000× *g* for 20 min at 4 °C. The supernatant was discarded, and the cell precipitate was placed in a fume hood to dry with a nitrogen blower. Then, a vacuum freeze dryer was used to obtain a white crystalline substance, which is the desired extraction sample.

Derivative: Derivatization was carried out with the derivatization reagent (2,4-dinitrofluorobenzene, DNFB) of ELITE. The dried sample was added with 200 μL of derivatization buffer and 100 μL of derivatized reagent. After mixing, the reaction was performed in a water bath at 60 °C for 1 h in the dark, and the equilibration buffer was added to the volume after cooling to room temperature. To 1 mL, stand at room temperature for 10 min, filter with 0.22 μL filter head, wrap in tin foil, and store at 4 °C in the dark, effective within one week.

Loading: Waters (Milford, MA, USA) HPLC instrument (2489 UV/Visible Detector, 1525 Binary HPLC pump) was used for detection, the maximum detection injection volume was 20 μL, the column was EliteAAK (5 μm, 4.6 × 250 mm), and the mobile phase was A: B = Phosphate buffer (2.7 mM KCl, 2.0 mM KH_2_PO_4_, 137 mM NaCl, 10 mM Na_2_HPO_4_, pH 7.4): 50% acetonitrile, The flow rate was 1.2 mL/min, the column oven was kept at 27 °C, the room temperature was kept at 20 °C, and the humidity was kept at 40%. The detection wavelength was 360 nm, the injection volume was 10 μL, and the total sample running time was 40 min.

Peak time: The peak time of Asp was 6.973 min, the peak time of Glu was 8.317 min, the peak time of Ser was 14.649 min, the peak time of DNP-OH was 14.315 min, the peak time of Arg was 16.380 min, the peak time of Gly was 16.748 min, and the peak time of Thr Time 17.219 min, Pro peak time 18.739 min, Ala peak time 19.389 min, Val peak time 24.253 min, Met peak time 25.554 min, Cys peak time 26.252 min, Ile peak time 27.380 min, The peak time of Leu was 28.021 min, the peak time of Trp was 30.537 min, the peak time of Phe was 31.113 min, the peak time of His was 31.380 min, the peak time of DNFB was 36.194 min, the peak time of Lys was 36.495 min, and the peak time of Tyr was 37.935 min.

### 4.9. Statistical Analysis

One-way ANOVA and Student’s *t*-tests were performed using GraphPad Prism (version 8.3.0, Boston, MA, USA). Student’s *t*-tests was used to compare Day 0 and Day 2, 3 M vs. 22 M. One-way ANOVA was used to compare the experimental group (si-SLC7A2-1, si-SLC7A2-2) with the control group (si-Control) and between different differentiation days, compare the mean of each column with the mean of a control column, correct for multiple comparisons using statistical hypothesis Dunnett testing. Statistical significance was defined as * *p* < 0.05, ** *p* < 0.01, *** *p* < 0.001, **** *p* < 0.0001. *p* < 0.05 was considered statistically significant.

## 5. Conclusions

In conclusion, our study revealed significant changes in the levels of amino acids during the differentiation of C2C12 myoblasts. We identified *Slc7a2* as a key gene encoding the arginine transporter, which was significantly upregulated during differentiation. Knockdown of *Slc7a2* led to decreased expression of important myogenic regulatory factors, resulting in impaired myoblast proliferation and differentiation. Our findings highlight the dynamic changes in amino acids and their transporters during muscle differentiation and provide insights into their functional roles. This study represents a comprehensive investigation in this field.

## Figures and Tables

**Figure 1 ijms-25-00095-f001:**
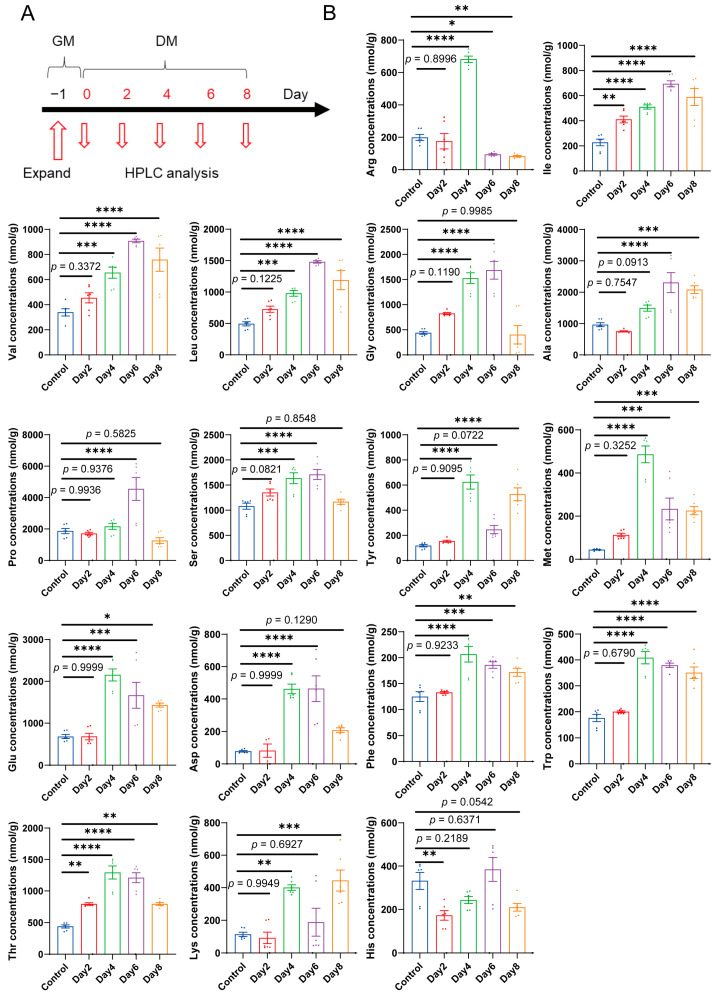
Dynamic change of amino acid levels in C2C12 myoblast differentiation. (**A**) The time course of C2C12 cells were sampled at 0, 2, 4, 6 and 8 days after differentiation. (**B**) During 0, 2, 4, 6, and 8 days of differentiation of C2C12 myoblast, the changes of different amino acids were detected by HPLC. (*n* = 6). The bars represent the mean ± SD. Statistical significance was determined by one-way ANOVA, comparing the mean of each column with the mean of a control column, correct for multiple comparisons using statistical hypothesis Dunnett testing. *, *p* < 0.05 vs. the si-Control.; **, *p* < 0.01 vs. the si-Control.; ***, *p* < 0.001 vs. the si-Control.; ****, *p* < 0.0001 vs. the si-Control.

**Figure 2 ijms-25-00095-f002:**
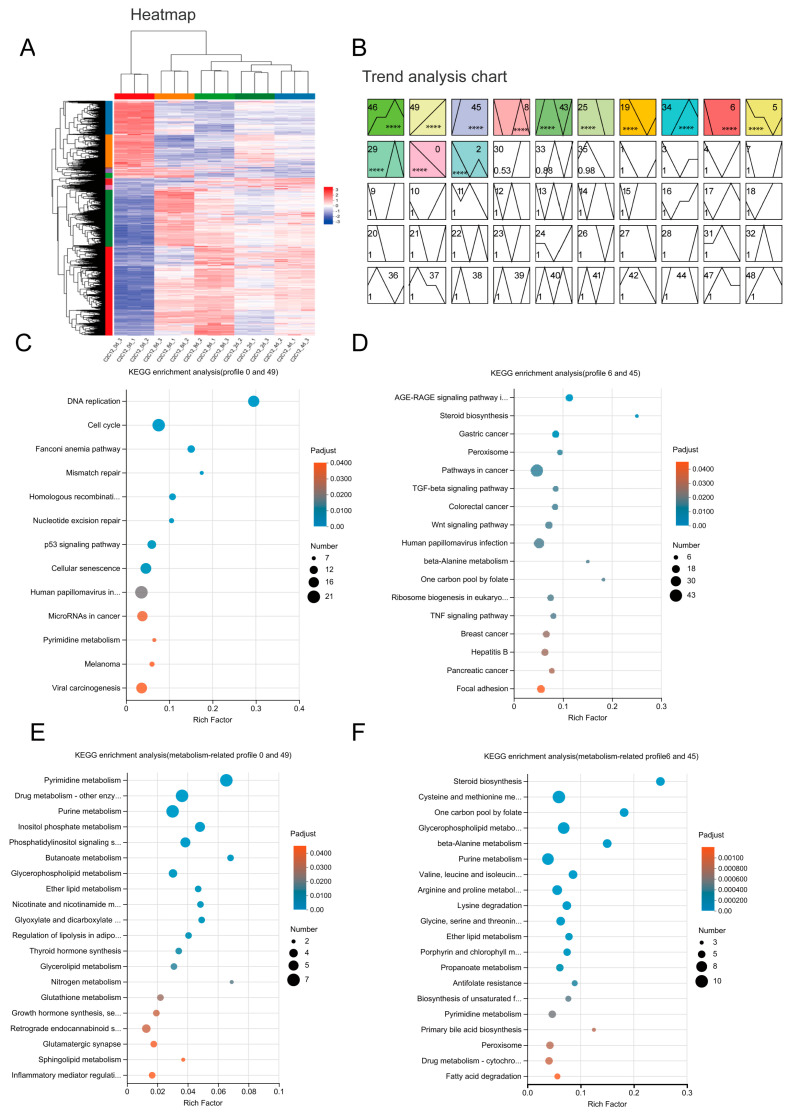
Gene expression analysis during C2C12 myoblast differentiation. (**A**) Heat map of row (gene) and column (sample) clustering. (**B**) Trend analysis of C2C12 cells at different differentiation times. Different colors represent different modules, ****, *p* < 0.0001 (**C**) KEGG enrichment of modules 0 and 49. (**D**) KEGG enrichment of modules 6 and 45. (**E**) Metabolism-related KEGG enrichment analysis of modules 0 and 49. (**F**) Metabolism-related KEGG enrichment analysis of modules 6 and 45.

**Figure 3 ijms-25-00095-f003:**
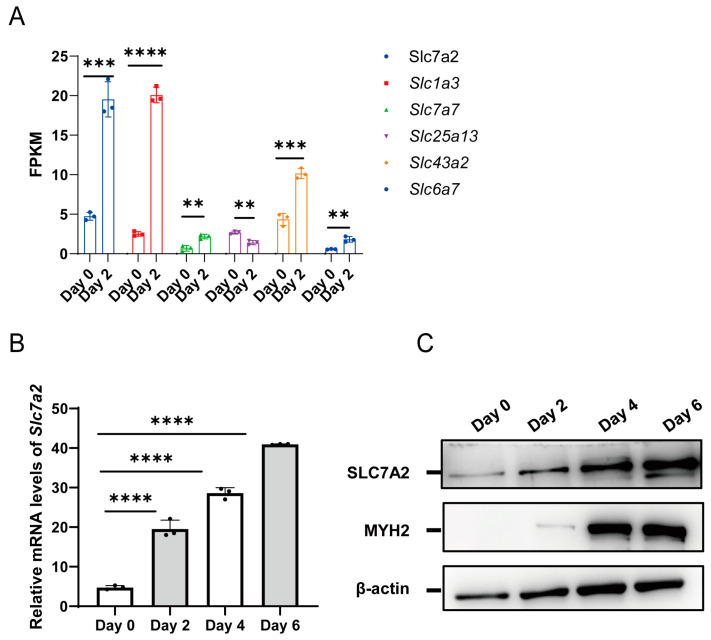
The expression level of SLC7A2 during myogenic differentiation. (**A**) The FPKM values of 6 amino acid transporter genes encoding C2C12 on day 0 and day 2 of myoblast differentiation were analyzed using RNA-seq data. (**B**) The mRNA levels of *Slc7a2* during myogenic differentiation. (*n* = 3) (**C**) Western blot analysis of protein levels of SLC7A2 during myogenic differentiation. Statistical significance was determined using one-way ANOVA. The bars represent the mean ± SD. (*n* = 3) **, *p* < 0.01 vs. the Day 0.; ***, *p* < 0.001 vs. the Day 0.; ****, *p* < 0.0001 vs. the Day 0.

**Figure 4 ijms-25-00095-f004:**
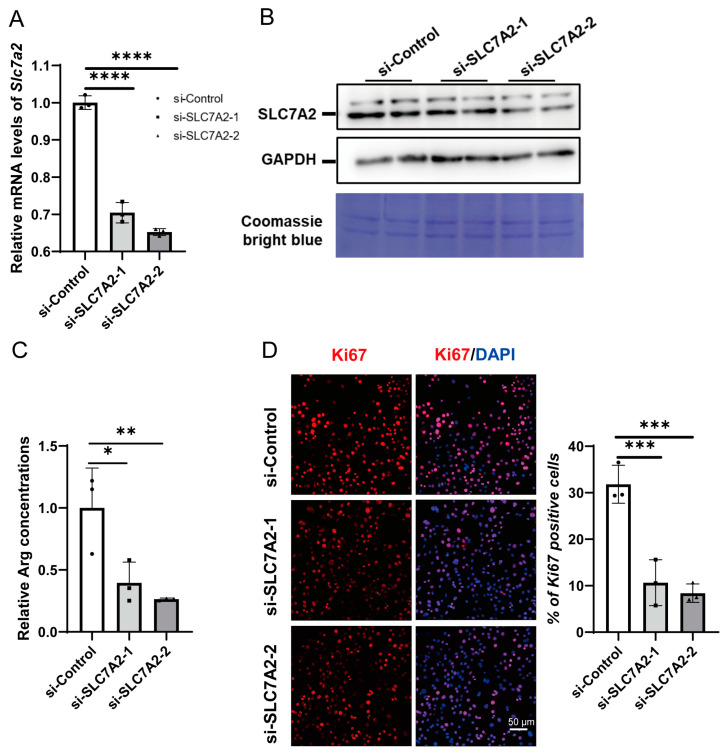
The effects of SLC7A2 knockdown in C2C12 myoblast proliferation. (**A**) The mRNA level of SLC7A2 in *Slc7a2* siRNA transfected myoblasts was detected using qPCR analysis (*n* = 3). (**B**) The protein level of SLC7A2 in *Slc7a2* siRNA transfected myoblasts was detected using western blotting (*n* = 3). (**C**) The relative Arg concentrations in *Slc7a2* siRNA transfected myoblasts (*n* = 3). (**D**) Immunocytochemistry for Ki67 (red) and DAPI (blue) in *Slc7a2* siRNA transfected myoblasts. The bars represent the mean ± SD. The bar graphs depicted the average foci number per cell of three independent experiments. There were 3 biological replicates per group. *, *p* < 0.05 vs. the si-Control.; **, *p* < 0.01 vs. the si-Control.; ***, *p* < 0.001 vs. the si-Control.; ****, *p* < 0.0001 vs. the si-Control. Statistical significance determined using one-way ANOVA.

**Figure 5 ijms-25-00095-f005:**
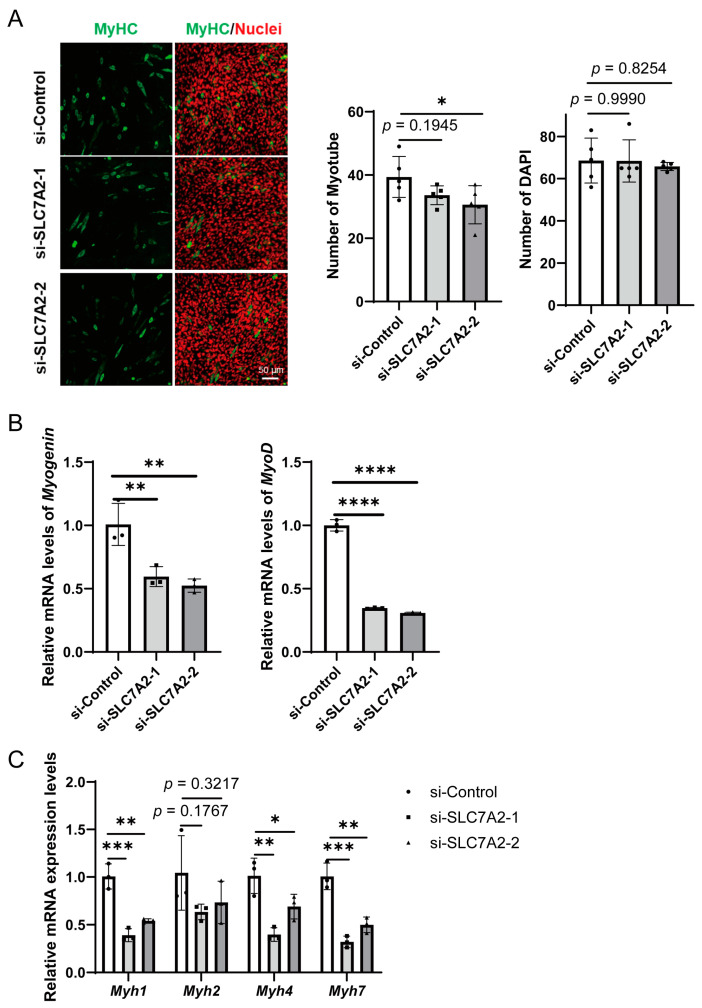
The effects of SLC7A2 knockdown in C2C12 myotube proliferation. (**A**) Immunocytochemistry for myosin heavy chain (MyHC; green) and nuclei (DAPI: red) in *Slc7a2* siRNA transfected myoblasts. The bars represent the mean ± SD. There were 5 biological replicates per group. (**B**) The mRNA levels of *Myogenin* and *MyoD* in *Slc7a2* siRNA transfected myotubes were detected using qPCR (*n* = 3). (**C**) The mRNA levels of *Myh7*, *Myh2*, *Myh1* and *Myh4* in *Slc7a2* siRNA transfected myotubes was detected using qPCR (*n* = 3). The bars represent the mean ± SD. *, *p* < 0.05 vs. the si-Control.; **, *p* < 0.01 vs. the si-Control.; ***, *p* < 0.001 vs. the si-Control.; ****, *p* < 0.0001 vs. the si-Control. Statistical significance was determined using one-way ANOVA.

**Figure 6 ijms-25-00095-f006:**
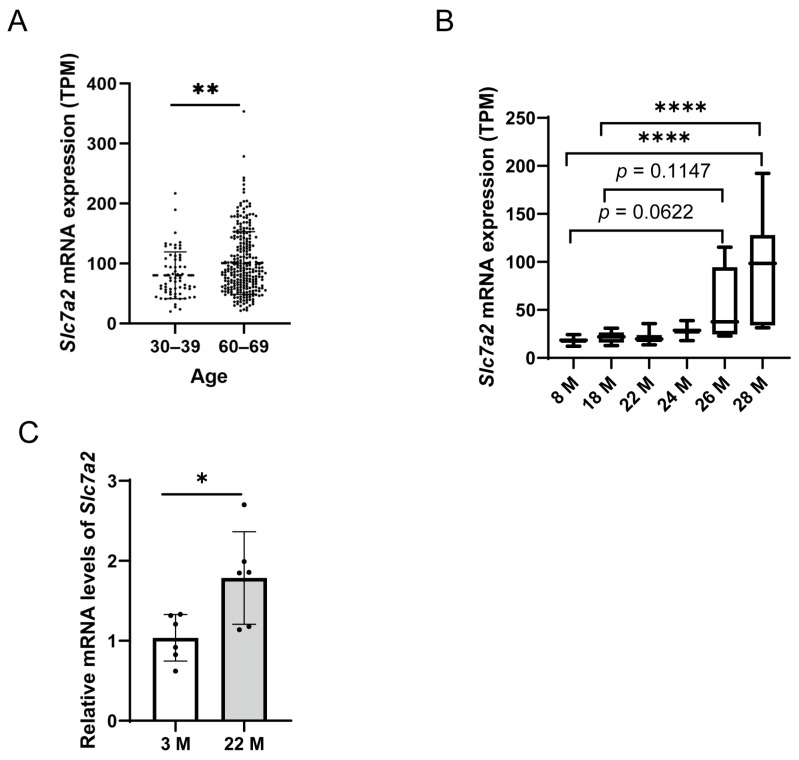
The expression of SLC7A2 in human and mice skeletal muscle during aging. (**A**) The expression of *Slc7a2* in young (30–39) and old (60–69)age groups of human skeletal muscle, gender: male and female. *p*-values were analyzed by Student’s *t*-test. **, *p* < 0.01 vs. the young (30–39). (**B**) The expression of *Slc7a2* in mice skeletal muscle at various ages. Statistical significance determined using one-way ANOVA, compare the mean of each column with the mean of every other column, correct for multiple comparisons using statistical hypothesis Tukey testing. ****, *p* < 0.0001. (**C**) The mRNA level of *Slc7a2* in gastrocnemius of young (3 month) and aged mice (22 month) were detected by qPCR, *p*-values were analyzed by Student’s *t*-test, (*n* = 6) *, *p* < 0.05 vs. 3 M. The bars represent the mean ± SD.

**Table 1 ijms-25-00095-t001:** Primers used in this study.

Gene Name	Primer Sequence (Forward)	Primer Sequence (Reverse)
*Slc7a2*	5′-GAGTAAGAGGCAGTCACCCG-3′	5′-ACTCGCTCTTCAAAGTCGCA-3′
*Myogenin*	5′-AATGGATTTGGACGCATTGGT-3′	5′-TTTGCACTGGTACGTGTTGAT-3′
*MyoD*	5′-GCACTACAGTGGCGACTCAGAT-3′	5′-TAGTAGGCGGTGTCGTAGCCAT-3′
*Myh1*	5′-GCGAATCGAGGCTCAGAACAA-3′	5′-GTAGTTCCGCCTTCGGTCTTG-3′
*Myh2*	5′-CAGAGGCAAGTAGTGGTGGA-3′	5′-CAAATTCTCTCTGAACAGGGCA-3′
*Myh4*	5′-ACACAGAGTCAGGCGAGTTT-3′	5′-CAGTGCGTTCTTGGCCTT-3′
*Myh7*	5′-CTTGCTACCCTCAGGTGGCT-3′	5′-GAGCCTTGGATTCTCAAACG-3′
*Gapdh*	5′-TGTCAAGCTCATTTCCTGGT-3′	5′-TAGGGCCTCTCTTGCTCAGT-3′

## Data Availability

The raw data generated in RNA-seq study were submitted to the CNCB database (accessed on 16 March 2023) under accession number CRA010184 (release date: 13 November 2023). Available online: https://ngdc.cncb.ac.cn/gsa/browse/CRA010184 (accessed on 16 October 2023).

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
