# Peer review of "Integrated Amino Acids and Transcriptome Analysis Reveals Arginine Transporter SLC7A2 Is a Novel Regulator of Myogenic Differentiation"

_ijms, 2023, doi:10.3390/ijms25010095_

Round 1

Reviewer 1 Report

Comments and Suggestions for Authors

This manuscript by Huang et al. has a stated objective to investigate the role of amino acids in skeletal muscle differentiation. The authors use C2C12 myoblasts as their model. The measures of cellular amino acids are interesting and the RNA sequencing is useful. Several points remain to be clarified and improved to make clear conclusions from this data set.

Major

1.     It needs to be shown whether the knockdown of SLC7A2 has any effect on cellular amino acids. The major objective of this study is to investigate the role of amino acids and amino acid transporters in skeletal muscle differentiation. Without knowing whether amino acid levels change after SLC7A2 knockout, it is not possible to conclude the role of amino acids in differentiation.

2.     Figure 4B. More information must be provided for the western blots, either in the figure, figure legend, or Methods. How was total protein measured? How much total protein was loaded per well. It is not sufficient just to “normalize” to intensity of the GAPDH protein band because GAPDH is incredibly abundant and has a very limited dynamic range (e.g. PMC3840294). Likely it is overloaded here.

3.     Figure 5A. What are the individual points in this plot? Are these from individual wells, or are these different microscopic fields? Plotting different fields from the same well is not appropriate because the different fields are not independent observations. Common practice is to quantify 5-10 fields per well, and then use the mean of those 5-10 fields as a single observation. Clarify how this was done and verify that plots and statistics are only done with independent observations, i.e. different wells.

4.     Figure 5A.  The number of nuclei should be counted and reported. Initial C2C12 differentiation rate is highly dependent on myoblast number/density. If proliferation is affected (as suggest by Fig 4), then it needs to be confirmed that myoblast number is similar across conditions.

5.     It is not clear how many replicates were done for each experiment. This should be given in the figure legend for each experiment.

Minor

6.     Line 122. “… highest of lowest ...”  Should this be “… highest or lowest …”

7.     Line 151-156. The logic is not clear. Since myoblast differentiation is extremely low in healthy/normal adult muscle, why would high expression levels of a particular protein in adult muscle indicate anything about myoblast differentiation? Clarify the rationale for selecting Slc7a2.

8.     Line 356.  “… added 3 ml to stop …”  What was added?

9.     Line 363. “The supernatant was discarded, and the supernatant was placed in …” How can the supernatant be discarded and also used for something else. Clarify this.

10.  Line 366. What is the derivatization reagent?

11.  Line 372. Add more details on the UPLC instrument. Waters produces many different instruments. What was the detector? The pumps? Was there an auto-injector?

12.  Line 374. Define the phosphate buffer.

13.  Line 378-386. How were peak times determined?

14.  Line 4.1. Were antibiotics used in the C2C12 work? Penicillin and streptomycin are common when using this cell line. If they were, state what concentration was used. If antibiotics weren’t used, state explicitly that antibiotics were not used.  

15.  Figure 1. The values are labeled as nmol/g. Is this grams of protein? Grams of total cell weight?  Clarify this either in the figure legend or in Methods.

16. Figure 5. The y-axis labels are confusing. Currently they read “Relation mRNA lever of …”  Should these be “Relative mRNA levels of …”?

Comments on the Quality of English Language

Fine.

Author Response

Thanks for your valuable comments and suggestions. We have revised our manuscript carefully according to your suggestions. See the attachment for details.

Reviewer 2 Report

Comments and Suggestions for Authors

The manuscript of Huang et al. presents an analysis of amino acid levels during myoblast differentiation and highlights their potential role during this process. In the introduction the process of myogenesis is described shortly and the metabolic pathways involved are characterized. Further, aminoacid trasporter families: SLC7, SLC1, and SLC6 are described in more detail.

The amino acid levels during different days in myoblasts diffrentiation were analysed in the first line with HPLC. In general, the peak of the aminoacid levels occurred at 4-6 days after differentiation and later on dropped. Further, RNA-Seq has been performed to determine the myoblasts expression profile, and enrichment pathways analysis have been done.

In the second part, the authors focused on the arginine transporter SLC7A2-one of the highest upregulated genes in expression analysis, highly expressed in muscle. SLC7A2 was analyzed during various differentiation days using Western Blot and mRNA analysis. Knockdown with Slc7a2 siRNA diminished the proliferation of myoblasts.  Also, the expression of SLC7A2 was upregulated during aging.

The manuscript is written clearly. Figures are of the good quality and illustrative.

Comments:

Line 85-86: first sentence is redundant

Figure 4 C- pictures in the left column are quite dark

In the discussion more reference to the literature would be helpful. Currently, the disussion summarizes mainly what have bene done. For example,  https://doi.org/10.1186/1471-2164-7-320 and https://doi.org/10.1371/journal.pbio.2005886 have not been cited at all

295-297 please chack if this sentence is needed

Paragraph 4.9 What statistical programs have been used?

Discussion ends with a conclusion and later there is a separate conclusion paragraph. Please avoid duplications.

Author Response

Thank you for your valuable comments and suggestions. We have carefully revised our original draft according to your suggestion. See the attachment for details.

Round 2

Reviewer 1 Report

Comments and Suggestions for Authors

The authors have done a very nice job responding to my initial review, and have shown what changes they would make in their point-by-point Response to Reviewers cover letter. However, they have not made the changes in the uploaded v2 manuscript. Please add the changes proposed directly into the manuscript.

Comments on the Quality of English Language

This is generally fine.

Author Response

We upload the revised manuscript, please review this manuscript, thank you!

Round 3

Reviewer 1 Report

Comments and Suggestions for Authors

Thank you. This looks great.

Comments on the Quality of English Language

Very good.

Author Response

Thanks for your valuable comments and suggestions. We have revised our manuscript carefully according to your suggestions. The detailed information is listed below.
